# Research Progress of Aluminum Phosphate Adjuvants and Their Action Mechanisms

**DOI:** 10.3390/pharmaceutics15061756

**Published:** 2023-06-17

**Authors:** Ting Zhang, Peng He, Dejia Guo, Kaixi Chen, Zhongyu Hu, Yening Zou

**Affiliations:** 1Sinovac Biotech Sciences Co., Ltd., Beijing 102601, China; zhangting@sinovac.com; 2Division of Hepatitis Virus & Enterovirus Vaccines, Key Laboratory of the Ministry of Health for Research on Quality and Standardization of Biotech Products, National Institutes for Food and Drug Control, Beijing 102619, China; hepeng9553@163.com; 3Sinovac Life Sciences Co., Ltd., Beijing 102601, China; guodj8210@sinovac.com (D.G.); chenkq7819@sinovac.com (K.C.)

**Keywords:** aluminum phosphate adjuvants, immune stimulation mechanism, aluminum phosphate improvement, preclinical and clinical

## Abstract

Although hundreds of different adjuvants have been tried, aluminum-containing adjuvants are by far the most widely used currently. It is worth mentioning that although aluminum-containing adjuvants have been commonly applied in vaccine production, their acting mechanism remains not completely clear. Thus far, researchers have proposed the following mechanisms: (1) depot effect, (2) phagocytosis, (3) activation of pro-inflammatory signaling pathway NLRP3, (4) host cell DNA release, and other mechanisms of action. Having an overview on recent studies to increase our comprehension on the mechanisms by which aluminum-containing adjuvants adsorb antigens and the effects of adsorption on antigen stability and immune response has become a mainstream research trend. Aluminum-containing adjuvants can enhance immune response through a variety of molecular pathways, but there are still significant challenges in designing effective immune-stimulating vaccine delivery systems with aluminum-containing adjuvants. At present, studies on the acting mechanism of aluminum-containing adjuvants mainly focus on aluminum hydroxide adjuvants. This review will take aluminum phosphate as a representative to discuss the immune stimulation mechanism of aluminum phosphate adjuvants and the differences between aluminum phosphate adjuvants and aluminum hydroxide adjuvants, as well as the research progress on the improvement of aluminum phosphate adjuvants (including the improvement of the adjuvant formula, nano-aluminum phosphate adjuvants and a first-grade composite adjuvant containing aluminum phosphate). Based on such related knowledge, determining optimal formulation to develop effective and safe aluminium-containing adjuvants for different vaccines will become more substantiated.

## 1. Introduction

Adjuvant, also known as immune regulator or immunepotentiator, is an additive to vaccines, most of which are not inherently antigenic. The adjuvants, belonging to non-specific immune enhancers, can enhance or alter the type of immune response to an antigen [1]. Adjuvant can not only assist antigen in inducing long-term and efficient specific immune responses in the body, so as to implement higher efficacy of vaccine and extend the protective time of immune response, but can also reduce the amount of antigen used, the costs of production and the times of immunizations. The purpose of using an ideal adjuvant is not only limited to enhancing immune response, but moreover enabling the body to obtain the best protective immunity. In 1920, Ramon of the Pasteur Institute in France found that abscesses at the injection site helped increase the titer of specific antibodies. To verify this hypothesis, he added substances such as starch or bread crumbs to the inactivated toxin and found that substances that induced inflammation at the injection site can increase the production of antibodies. In 1926, Glenny et al. first reported the immunological enhancement of diphtheria toxoid precipitated by insoluble aluminum salts [2], laying the foundation for the widespread use of aluminum-containing adjuvants (such as aluminum hydroxide and aluminum phosphate). Since the introduction of the concept of adjuvants, interests in vaccine adjuvants have increased, and the significance and importance of adjuvant research have also become increasingly prominent. According to their chemical properties, adjuvants can be roughly divided into the following categories: inorganic adjuvants such as aluminum-containing adjuvants, emulsion-type adjuvants such as MF59 and AS03, water soluble adjuvants such as saponins, adjuvants targeting pattern recognition receptors such as MPL, and cytogenic adjuvants such as interleukins (ILs). The combination of adjuvants with antigens can increase the surface area of antigens [3], and its biological effects mainly include: (1) antigen depot effect [4], (2) up-regulation of cytokine and chemokine expression, recruitment of immune cells at the injection site [5], (3) activation of inflammasome [6], and (4) vector effect (promoting activation and maturation of dendritic cells, enhancing antigen presentation) [7]. On one hand, with the adjuvant effect, antigens are more likely to be phagocytized by macrophages, and are processed and presented effectively [8]; on the other hand, adjuvants can change the physical properties of antigens, which results in slower release of antigens in the body and prolonged interaction time between antigens and immune cells [9,10]. At present, hundreds of natural or synthetic compounds have been used in the research of adjuvants, but the number of approved adjuvants for vaccines is still limited, except for aluminum salt adjuvants, only MF59, AS04, AS03, AS01, CpG1018 and Matrix-M for COVID-19 emergency use are approved and marketed [11]. MF59 and AS03, as new oil-in-water emulsion adjuvants, are widely used in influenza vaccines, such as H1N1 and H5N1. MF59 can induce humoral immunity without potential toxicity [12]. AS03 can improve the local innate immune response profile and have a stronger antigen–antibody specific response, but the natural sources of squalene are limited due to its complex operation steps and low yield. Saponin adjuvants such as Quil-A and QS-21 can simultaneously stimulate Th1 and Th2 immune responses [13]. QS-21 is used as part of its adjuvant system in both the GSK malaria vaccine (Mosquirx) [14] and the herpes zoster vaccine (Shingrix) [15]. However, the inherent shortcomings of QS-21, including its scarcity, heterogeneity, hydrolysis instability and dose-limiting toxicity, have limited its clinical progress as a stand-alone adjuvant. The hemolytic effect and hydrolysis instability of saponin compounds can be solved by built-in liposomes or ISCOMS. However, its limited supply has limited its wide application in products [16,17]. CpG adjuvants are synthetic single-stranded DNA containing one or more unmethylated CPG motifs. The CpG 1018 adjuvant used in the Heplisav-B vaccine has good safety and strong immune memory [18]. However, when co-administered with antigens, only Th1-biased immune responses were induced, and only some antigens had immune enhancement [19]. Aluminum hydroxide (Al(OH)_3_, AH) and aluminum phosphate (AlPO_4_, AP) still dominate the field of adjuvants for the formulations of human vaccines, list of licensed type of adjuvant and aluminum content of aluminum-adjuvanted vaccines for use in the United States are summarized in Table 1.

Although aluminum-containing adjuvants have been applied nearly several decades in purpose of enhancing vaccine immune response, the action of aluminum-containing adjuvants still cannot be explained thoroughly, especially the molecular mechanism. Theoretical basis for adjuvant type selection is still lacking, and the effect of adjuvants on vaccine antigen formulations cannot be accurately predicted and must be determined through clinical trials. Continuing to use aluminum-containing adjuvants also requires us to understand their complexity and acquire the expertise needed for vaccine formulation optimization. Flarend et al. [20] used 26Al isotope as a tracer to study the pharmacokinetics of aluminum after intramuscular injection of AH and AP (0.85 mg/dose). As data showing, 17% AH and 51% AP were released into the circulation 28 days after injection. Based on the mentioned results and other estimates from publicly published data, the concentration of aluminium in the blood of infants during their first year of life as a result of vaccination remains well below UNICEF’s minimum risk levels. In accordance of the long-term success of aluminum-containing adjuvants, researchers are still convincing that aluminum-containing adjuvants should be continuously considered as the “gold standard” for all adjuvants at least up to now. New vaccine candidates need to be evaluated with aluminum-containing adjuvants before considering other novel adjuvants if they require the adjuvants to induce a protective immune response. Recent studies have significantly improved our understanding of the physical, chemical, and biological properties of these adjuvants, providing a critical theoretical basis for further understanding of their underlying mechanisms. Seeber et al. [21] found that the dissolution rate of aluminum phosphate adjuvant in simulated interstitial fluid was significantly higher than that of aluminum hydroxide adjuvant. The dissolution pattern of aluminum-containing adjuvants in the tissue fluid varies greatly and may affect the antigen release pattern. By comparing the in vitro and in vivo innate immune response induced by AH and AP, and determining the immune pathways activated by these two adjuvants through proteomic analysis of human primary monocytes, Kooijman et al. [22] revealed that AH and AP have different regulatory effects in different immune system-related processes. The previously published articles used AH to study the mechanism of aluminum-containing adjuvants in enhancing immune response but the purpose of this article is to summarize the existing research on the mechanism of AP, and the current development status of AP, in order to provide a theoretical basis for determining the quality attributes of each adjuvant and selecting the appropriate type of adjuvant in the clinical development stage as early as possible.

## 2. AP

AP has been shown to be a safe and effective adjuvant. Aluminum phosphate adjuvant has no fixed molecular formula, and its approximate molecular formula is [Al(OH)x(PO_4_)y]. X-ray test results of AP show diffraction band without crystal characteristics, indicating its amorphous structure [1,23]. Transmission electron microscopy shows that AP is a plate particle network, and the primary particle is a disk structure with a diameter of 50 nm, which forms suspended particles with the diameter of approximately 3 μm. The preparation of AP is by mixing aluminum salt solution (usually AlCl_3_ or KAl(SO_4_)_2_) with trisodium phosphate alkaline solution, and then precipitating with sodium hydroxide [24]. Aluminum phosphate is a compound of hydroxy-aluminum phosphate, similar to aluminum hydroxide, and the hydroxyl group on its surface can also lose or gain protons under different pH conditions, thus changing the surface charge. The degree of replacement of hydroxyl groups by phosphate groups depends on the reactant and precipitation conditions and their isoelectric point (PZC); thus, changing the solution environment can lead to the desorption or adsorption of phosphate on the surface of aluminum adjuvant, accompanied by the change of surface charge and antigen/adjuvant interaction [25]. There is no fixed ratio between hydroxyl and phosphate, so the change on pH during precipitation will affect the ratio of hydroxyl to phosphate groups, and the change on the phosphorus to aluminum (P/Al) ratio will result in PZC values ranging from five to seven. Typical commercial AP has P/Al of 1.1 to 1.15 and a PZC of 4.6 to 5.6. At higher pH, the increase in hydroxyl concentration can make the hydroxyl group compete more effectively for the coordination sites around aluminum, while at lower pH, precipitation is more conducive to the formation of Al-PO_4_ bond rather than Al-OH bond [26]. The resulting differences in groups will further lead to aggregation changes, thus affecting the particle size. Additionally, the exposure of different groups on the surface of the adjuvant will also lead to a great difference in the type and ability of protein adsorption. Studies have shown that some antigens can be well-adsorbed to the surface of aluminum adjuvant under the condition of pH below 6. If the pH value deviates from this range, additional ions (such as phosphoric acid groups, amino acids, peptides, polysaccharides and other impurities) compete with the antigen for adsorption sites, thus reducing the antigen adsorption. In general, low ionic strength, few phosphoric acid groups, and few impurities facilitate antigen adsorption to the surface of the aluminum adjuvant.

## 3. AP Binding to Antigen

### 3.1. Mechanism of AP Binding to Antigen

Antigens can usually be adsorbed to aluminum-containing adjuvants through hydrogen bonding, van der Waals force, hydrophobic action, electrostatic attraction and ligand exchange [23,27], among which electrostatic attraction is the most common. When opposite charges separately belong to adjuvants and antigens, the adsorption usually depends on electrostatic attraction. Under neutral conditions, AP can adsorb positively charged antigens [28]. Jones et al. [8] found that the adsorption of AlPO_4_ by lysozyme was mainly driven by electrostatic attraction between positively charged lysozyme and AP. At pH 3, known as an acidic environment, a positive charge is caused for both lysozyme and AlPO_4_, significantly increasing their surface charges compared to their conditions at pH 9; however, the charge complementarity between the two entities is predicted as maximum at pH 9. At pH 6, when increasing the amount of lysozyme from 0.25 to 2 mg mL^−1^, the net surface charge has no significant change [27]. Ligand exchange is the strongest adsorption between antigen and adjuvant, which can happen even when the adjuvant and antigen have opposite charges [29]. When the surfactant was present in the vaccine formulation, the hydrophobic action between the antigen and adjuvant increased and the electrostatic adsorption between the antigen and adjuvant decreased. These forces are usually necessary to induce a protective immune response against recombinant subunit antigens and proteotoxins, and appropriate adjuvants need to be screened according to the characteristics of the antigens. The earliest vaccine using AP was pertussis vaccine, which was applied in 1942 [30]. Up to now, AP is still a widely used adjuvant, which has been formulated in a variety of vaccines, including diphtheria vaccine, tetanus vaccine, poliomyelitis vaccine, 13-valent pneumococcal conjugate vaccine, HPV vaccine, etc. Rosado et al. [31] evaluated AP as an adjuvant for DNA vaccines against Lactobacillus mexicana with different formulations and dosages in BALB/c mice, and found that AP is an effective, low-cost and safe adjuvant that can be used for DNA vaccines against intracellular pathogens. Ulmer et al. [32] compared DNA vaccines against botulinum toxin containing AH or AP, and found that AP notably improved the efficacy of DNA vaccine against botulinum toxin (BoNTs), while AH decreased the humoral immune response and protective efficacy of DNA vaccine against BoNTs. The properties of the two aluminum-containing adjuvants in DNA vaccines are significantly different, and their abilities to act as DNA vaccine adjuvants are inversely proportional to their abilities in DNA binding. When formulated in normal saline, the mechanism by which AP improves the efficacy of the DNA vaccine through the adjuvant effect is not by increasing the expression of antigenic proteins in vivo, but by acting on the antigen after the expression of the DNA vaccine and acting as an adjuvant on the antigen protein [33]. Therefore, the authors speculate that DNA vaccines may need to keep most of the plasmid DNA un-adsorbed by AP adjuvants in order to express antigenic proteins effectively. The adsorption of the AH adjuvant on plasmid DNA reduces the expression of antigen proteins, which may reduce the immune effect of the DNA vaccine [34]. In addition, other studies have confirmed that aluminum phosphate can reduce the effective dose for inducing protective immune response against Lactobacillus mexicana of plasmid DNA, which is consistent with results of previous studies [35,36].

### 3.2. Analytical Characterization of Aluminum-Adsorbed Antigens

The physical and chemical properties of adjuvants are not sufficient to predict the efficacy and stability of vaccines containing aluminum adjuvants, therefore, quality control of aluminum adjuvant vaccines is necessary.

(1) Firstly, the surface charge properties of aluminum-adsorbed antigens need to be understood to predict the physical properties of vaccine suspensions. (2) Secondly, the determination of the adsorption rate and adsorption strength of aluminum-adsorbed antigens can help us to understand the adsorption capacity of aluminum adjuvants on antigens, as well as the adsorption capacity under different conditions (such as different pH or ionic strength) and long-term storage. (3) Determination of the dissociation kinetics of aluminum-adsorbed antigens can help us understand the rate of antigen dissociation from aluminum adjuvants and the stability of dissociation under different conditions (such as different temperatures or times). Gun et al. used antigen fingerprinting in an antigen-specific manner for adsorbed conjugate vaccine consistency testing without desorbing or otherwise pre-treating the final vaccine. In addition, aluminum adjuvants may affect the structure of the antigen and thus its immunogenicity. The structure of the antigen can be analyzed by methods such as X-ray crystallography or nuclear magnetic resonance (NMR), and the thermal stability of the antigen before and after adsorption can be determined by methods such as Differential Scanning Calorimetry (DSC) or Differential Scanning Fluorimetry (DSF). (4) Finally, the immunogenicity of aluminum-adsorbed antigens needs to be evaluated. This is usually achieved by immunization in an animal model and then measuring the antibody response, cellular immune response and other indicators in the animals. This can help us to understand the effect of aluminum-adsorbed antigens on the immune response after adsorption and the changes in the immune response under different conditions (such as different doses or vaccination intervals).

## 4. Immune Stimulation Mechanism of AP

### 4.1. Depot Effect

In most cases, the antigens used in vaccine products are weak in immunogenicity, which can be enhanced by binding to adjuvants. After adsorption, the antigen accumulates on the surface of and inside the adjuvant particles, helping the antigen maintain its physical and chemical properties [37]. A huge number of studies have indicated that antigens can be combined with adjuvants and released slowly at the injection site to continuously stimulate the immune system and enhance immune response, which is called the depot effect. Harrison first demonstrated the existence of the depot effect by inoculating aluminium-containing nodules from one guinea pig into another guinea pig [38]. Taking AP for example, the antigen is mixed with aluminum phosphate into a gel state. When the mixture is injected into the body, it creates a kind of “reservoir” in the body. These insoluble gel particles adsorb and disperse the antigenic components, increasing the surface area of the antigen and forming a granuloma at the injection site. Antigens in the granuloma will slowly penetrate into the body and preserve the antigens that can only stay at the injection site temporarily for several weeks, so that the antigens will not be cleared by the liver in a short time and can achieve the purpose of long-term stimulating the immune system [39]. The depot effect at the vaccine injection site has long been considered one of the main action mechanisms of immune adjuvants. The depot effect is influenced by the physical properties of aluminum-containing adjuvants (surface area, charge, and morphologic structure). For example, a larger surface area may enhance antigen adsorption, promote antigen storage, and facilitate antigen presentation to antigen-presenting cells (APCs). Ultimately, the immune response is enhanced [40]. Antigens that are bound to aluminum-containing adjuvants are not only beneficial for presenting to antigen-presenting cells, but can also slowly release into tissues along with the decomposition of aluminum-containing adjuvants, leading to the delay of antigens consumption and their stimulation time to the immune system can also be prolonged. In addition, the prolonged interaction interval between antigens and presenting cells also enhances the induced immune response [41].

The depot effect is an important mechanism by which adjuvants play their role, but the depot effect alone cannot fully explain the mechanism by which aluminum-containing adjuvants enhance immune stimulation. Some studies have found that the depot is not a necessary condition for the function of an aluminum adjuvant: for example, Holt et al. [42] found that when diphtheria toxin was absorbed on an aluminum adjuvant and injected into guinea pigs, even if the inoculated tissue was removed seven days after inoculation, the effect caused by vaccination still existed. Hutchison et al. [43] found that removal of the inoculation site two hours after inoculation of the antigen–aluminum adjuvant mixture had no effect on the specific antibodies and T cell response produced by the body. In addition, Gupta et al. mixed C^14^-labeled tetanus toxoid with aluminum adjuvant for injection, and found that the antigen was not “stored”, but immediately released by monitoring the trace of C^14^ [44]. These results indicate that although the depot effect is one of the important mechanisms of immune adjuvants, it is not a necessary mechanism for all adjuvants to function. However, in the case of the antigen depot effect, further studies are required to validate its significance.

### 4.2. Recruitment of Immune Cells

The immune system has three defense lines to protect human health: physical and chemical barriers, innate immunity, and adaptive immune responses. After intramuscularly injection, an aluminum adjuvant can stimulate the innate immunity of the body. The main participants of innate immunity include dendritic cells, macrophages, monocytes, neutrophils, eosinophils, basophils, mast cells, natural killer cells, interferon and complement proteins [45].

Both AH and AP activate immune system and immune system-related pathways in monocytes, and the in vitro immune response to AH is more pronounced compared to AP. At present, Kooijman et al. [22] proved through in vivo and in vitro experiments that AH and AP could recruit different kinds of cells and trigger different immune responses after injection. Because monocytes are the hub of innate immunity and adaptive immunity, the changes of monocytes are very important in studying the mechanism of adjuvant action. Both AH and AP can activate innate immunity in vivo. In vivo experiments show that only intramuscular immunity of AH can attract neutrophils, while AP can attract mononuclear cells and macrophages to the injection site [46]. These results indicate that AH and AP have different roles in innate immunity, which may be explained by that the different physical and chemical properties of these two adjuvants could cause different effects on cells. AH and AP adjuvanted antigens were injected into the muscle of mice, and proteomic analysis of the injection site showed that 67% of the up-regulated proteins near the injection site overlapped after the injection of the two aluminum-containing adjuvants, indicating that the two adjuvants had certain similarities in stimulating the immune response, compared with AP, AH induced more immune system-related pathways, and in vivo, more neutrophils were attracted to the injection site [22,47]. The recruitment of immune cells at the injection site provides favorable conditions for adaptive immunity.

### 4.3. Enhancement of Antigen Uptake by Antigen Presenting Cells (APCs)

A key step of inducing immune response is antigen uptake by APCs. Both the antigen adsorbated by aluminum adjuvant and the antigen free in the intercellular substance can be uptaken by APCs, but the former particles form can be taken up by APCs more easily [48]. The adsorption between antigens and adjuvants enables the antigen to maintain a high concentration at the injection site and release slowly, so as to prolong the time of antigen uptake by APCs and the effect of the antigen on the immune system [49].

Kooijman et al. [22] found that AH was superior to AP in aiding antigen presentation and processing. After 24 h incubation of AlPO_4_, antigen presentation and processing are down-regulated, and at this time, there is no down-regulation of AH incubation. This pathway can recruit neutrophils in AH adjuvanted antigens, but not in AP adjuvanted antigens. However, injection of antigens containing AP can recruit mononuclear/macrophage cells which are strongly up-regulated after 48 h incubation, and only with AP. This suggests some biological differences between AH and AP, and the choice of these adjuvants should be carefully considered in vaccine formulations. APCs include macrophages, B cells and dendritic cells (DCs). DCs can provide the widest range of exogenous antigens for T cells that play a key role in immune response. Only mature DCs can effectively activate T cells, and maturation of DCs is marked by the expression of co-stimulatory molecules such as CD80 and CD86 on the membrane surface [45]. DCs use protease in lysosomes and pH changes to process antigens into peptides, which are then expressed on cell membranes in the form of an antigen peptide-MHC Class II molecular complex and presented to CD4^+^ T cells to stimulate the differentiation of Th into Th2 [50]. An experiment performed by Sokolovska et al. showed that both AH- and AP-incubated CDs derived from mouse bone marrow could increase the expressions of CD80 and CD86 and promote the maturity of CDs. The effect of AH on increasing the expressions of CD80 and CD86 was significantly stronger than that of AP. These two adjuvants can also induce DCs to produce IL-1 and IL-18, while IL-1β and IL-18 can promote the differentiation of CD4^+^ T cells into Th2 cells as well [50,51].

### 4.4. Activation of NLRP3 Inflammasome Pro-Inflammatory Signaling Pathway

Aluminum-containing adjuvants can recruit leukocytes, promote differentiation of dendritic cells, and accelerate local tissue inflammation independent of Toll-like receptors. However, the cellular targets that release the pro-inflammatory activity of aluminum-containing adjuvants were not identified until recently. Data from various laboratories have shown that aluminum-containing adjuvants can target nucleotide-binding oligomerization domains (NOD), such as receptor protein 3 (NLRP3). NLRP3 is a member of the inflammasome, which can recognize danger signals transmitted into the cell. The main responsibility of macrophages is phagocytosis and processing antigens, and aluminum-containing adjuvants can induce and activate endogenous immune responses through NLRP3, thereby promoting the secretion of high levels pro-inflammatory cytokines, such as IL-1β and IL-18 from macrophages; however, they are not present when lacking NLRP3 inflammasome components in cells [6]. Kooijman et al. [22] found that aluminum adjuvants can participate in the innate and adaptive immunity induced by ovalbumin through activation of the NLRP3 inflammasome.

NLRP3 is one of the members of NOD-like receptor family. Dead cells, silica, asbestos, aluminum adjuvant and other components can stimulate and activate NLRP3 [52,53]. NLRP3 inflammasome oligomerization can be achieved by caspase activation and recruitment domain (CARD). CARD interacts with aspartic protease 1 to form inflammatory bodies. When NLRP3 is stimulated, it combines with apoptosis-related specks such as protein ASC and pro-caspase-1 to assemble NLRP3 inflammasome. The activated NLRP3 inflammasome prompts ACS to lyse pro-caspase-1 into caspase-1. Caspase-1 then promotes the activation of IL-1β and IL-18 [52,53], which in turn promote the differentiation of CD4^+^ T cells into Th2 cells, enhancing the immune response, and Th2 cells stimulate the production of antibodies by B cells through secretion of cytokines to form humoral immunity [45,54].

Intracellular lysosomes can phagocyte aluminum-containing adjuvants ingested by APCs, and cathepsin B released after that can promote the assembly of NLRP3 inflammasome [55]. In addition, AH and AP are toxic to cells, and the injection of the aluminum adjuvant vaccine will cause cell damage and release of cellular DNA, ATP and uric acid [56]. AP is more toxic to monocytic leukemia cells (THP-1 cells) than AH. When the adjuvant concentration was 100.0 μg/mL, the mortality rate of THP-1 cells incubated with AP was more than 50%, while the result of AH was about 20% [57,58]. DNA, uric acid and other intracellular molecules released after cell death can act as damage-associated molecular patterns (DAMPs) to further activate innate immune cells [53,59].

The inhibition of caspase-1 showed that both AH and AP could induce caspase-1 activation, and then promote the maturation of IL-1β and IL-18. Ruwona et al. also showed that both AH and AP can promote IL-1β secretion by stimulating the NLRP3 inflammatorome, but AH can stimulate THP-1 cells to produce higher levels of IL-1β than AP, which may be related to the differences in the structure, density, surface charge and charge density between these two aluminum-containing adjuvants [60].

### 4.5. Other Immune Stimulatory Mechanisms

Traditionally, aluminum adjuvants have been thought to primarily induce a cellular immune response that involves the activation and proliferation of T cells. However, the results of Scholl et al. found that the aluminum adjuvant that adsorbed rNanH and rPknG proteins from *Corynebacterium pseudotuberculosis* vaccine produced partial protection in animals. Significant levels of total IgG, IgG1, and IgG2a antibodies against *C. pseudotuberculosis* were also produced. The levels of IgG2a isotypes were higher, and the antibodies produced were characterized by a mixed humoral response with a tendency toward Th1 type [61]. Rezende et al. adsorbed acid phosphatase rCP01850 with AH, it induced Th1/Th2 mixed immune response, which improved the survival rate after challenge [62].

Ramanathan et al. [63] found in guinea pigs that AH could activate complement in an alternative way and induce granulation formation and macrophage damage. In addition, dendritic cells can recruit aluminum-adsorbed antigen-antibodies, and complement factors can regulate receptors on B cells to form aluminum-adsorbed antigen-antibodies, thereby promoting immune response. Thus, aluminum-containing adjuvants can activate complements and further enhance the immune response via B cells and dendritic cells [49,64]. Modes of action of aluminum-containing adjuvants are summarized in Figure 1.

## 5. Main Factors Affecting the Immunogenicity of AP Adjuvanted Vaccines

### 5.1. Adsorption Intensity

The intensity of antigen adsorption is a main factor affecting the immune response. The adsorption of AP can be evaluated with two parameters: maximum adsorption amount of a monolayer [2], characterized by the adsorption capacity; and adsorption intensity, characterized by the adsorption coefficient. Glenny et al. [2] observed that injection of alum-precipitated diphtheria toxoid caused a significant improvement on immune response in 1926. Furthermore, when the precipitate was filtered, the filtrate did not contain the toxoid. This result leads to the conclusion that the adsorption of the antigen into an aluminum-containing adjuvant before administration is quite important for immunogenicity enhancement. Studies to date have also focused on adsorption capacity, for the aim of complete antigen adsorption. Only a few researchers have investigated the adsorption coefficient [57], among which Hansen et al. [9] examined the relationship between adsorption coefficient and immune enhancement in their study. They prepared four vaccines with varied adsorption coefficients acquired by changing phosphate groups’ number on the antigen (α-casein and de-phosphorylated α-casein) or hydroxyl groups’ number on the adjuvant surface (AH and phosphate-treated AH) and found that aluminum hydroxide had the highest adsorption coefficient, which decreased when phosphate groups were added. When the four vaccines were exposed to tissue fluids or normal human plasma, the degrees of dissociation were negatively correlated with the adsorption coefficients in vitro, and the geometric mean titers of antibodies induced after immunization of mice were also inversely correlated with the adsorption coefficients. The activation of the T cell was not observed in mice immunized with the vaccine, which had a maximum adsorption coefficient (α-casein) for AH, suggesting that antigen processing and presentation to T cells is impaired when the adsorption to the antigen is too strong, thus affecting the immunogenicity of the antigen. These results provide an important reference for studying the effect of adsorption strength on aluminum adjuvant [3,65]. Chang et al. [59] also found that when an antigen and aluminum adjuvant were injected simultaneously, the antigen not adsorbed to the adjuvant can still stimulate an immune response. It is inconclusive whether there is a difference in the effect of antigen adsorption onto aluminum adjuvant on the intensity of antigen immunogenicity. In addition, it was found during aging at room temperature that AP became more ordered, leading to a reduction in its ability to adsorb onto proteins [60,66]. In general, the immunogenicity of the antigen increases after adsorption onto the aluminum adjuvant, but the stability of the antigen decreases as the adsorption time increases [67]. In addition, proteins adsorbed onto the surface of solid particles are prone to de-folding and losing their secondary and tertiary structures [68], thus exposing more hydrophobic groups and further enhancing their ability to bind to aluminum-containing adjuvants. Therefore, the adsorption strength of AP shows an essential influence on the immunogenicity, safety and stability of the antigen–AP complex.

### 5.2. P/Al

P/Al is the molar ratio of phosphorus and aluminum elements in AP. AP is prepared by replacing the hydroxyl groups on the surface of AH with phosphate, in which there is no fixed value of phosphorus to aluminum ratio, so the different ratios of phosphorus to aluminum in AP may show different influence on the immune response. Hansen et al. showed that the intensity of ligand exchange adsorption can be changed by pre-treating AH with phosphate to reduce the number of hydroxyl groups, and the immune response can be also optimized in this way [9]. Hem et al. [24] altered the surface charge properties of AH by changing the P/Al through replacing the hydroxyl groups on the surface of AH with phosphates. The substitution of higher valence anions reduces the zero charge point. Treatment with phosphate anions induces an increase in negative surface charge, and modification of the surface charge properties of AH by pretreatment of adjuvants with phosphate anions will result in the replacement of hydroxyl groups on the adjuvant surface by phosphates and a reduction in PZC. For positively charged proteins, the amorphous nature of these compounds build a larger surface area and higher adsorption capacity.

### 5.3. PH and Ionic Strength

Vaccine suspensions may transition between flocculation and deflocculation states, and pH can directly affect the sedimentation behavior of vaccine suspensions with aluminum phosphate. Muthurania et al. [69] showed that the pH of AlPO_4_ has a significant effect on its surface charge. The PZC of AlPO_4_ is between 5 and 7. AlPO_4_ shows a positive charge at low pH and a negative charge at high pH. At high zeta potential, there are mainly repulsive forces between the particles, which are completely dispersed in the medium, and the particles exist as separate entities in the suspension, where the particles settle at different rates due to the wide distribution of particle size and the slow settling rate. Deposits formed in this way usually accumulate closely together, forming hard lumps that are difficult to re-disperse [70]. Adding oppositely charged ions or changing the solution pH can lead to a decrease in the magnitude of the zeta potential; thus, particles tend to clump together more tightly, forming loose aggregates or flocs due to the repulsion decrease. Systems that exhibit this property are called flocculation. By increasing the ionic intensity of the formulation, the concentration of sodium chloride (NaCl) from 0 mM to 1000 mM also impacts the charge state of AlPO_4_ particles and the colloidal interactions between particles, showing a decrease trend in the slope (and variation in magnitude) related to a decrease in zeta potential at the two extreme pH values. This is due to the shielding effect of NaCl, i.e., NaCl reduces the electrical potential between AlPO_4_ particles and water molecules in solutions, which leads to the observed decrease in zeta potential. This in turn results in an increase in the PZC of AlPO_4_ from approximately 4.7 when NaCl is absent to approximately 6.4 when containing 1000 mM NaCl. The salt ion modulates the types of colloidal interactions present in the formulation (including electrostatic interactions, van der Waals interactions or hydrophobic interactions) present in the formulation [71]. Thus, the pH and ionic strength of the solution significantly modify the settling suspension characteristics of the adjuvant and adsorption product mixture, which can shift between flocculated and non-flocculated states. These particles exist as loose aggregates, with higher sedimenting rates when they are clustered together as a population. Sediments formed in this way are looser and usually have a scaffold-like structure that is prone to resuspension [72]. Thus far, there is a lack of clear understanding of the relationship between the suspension behavior of aluminum-containing adjuvant vaccine mixtures and their immunogenicity [73]. However, it has been validated that the sedimentation behavior severely affects the dispersion state of the mixture, thus further affecting the safety and stability of the vaccine.

### 5.4. Particle Size

The two main factors governing the performance of adjuvant suspensions are particle size and the charge of the dispersed particles. A variety of ways can be used to control particle size, such as shear or adding surfactants or excipients, while under given solution conditions or antigen adsorption, the existence of favorable or unfavorable particle interactions are determined by the latter way. It is well-known that particle size affects the settling rate in a suspension and can be described optimally using the Stokes equation, in which the settling rate of particles is proportional to the square of the diameters of dispersed particles in the suspension. However, the limitation of gravitational sedimentation is always given by the particles number and size [72]. At low concentrations, particles can settle freely without interfering with each other by collisions and/or Brownian motion [70]. Adding ions with opposite charges or changing the solution pH results in a decrease in the magnitude of the zeta potential. When giving a certain pH or adding a certain concentration of ions, due to the decrease in repulsive forces, the particles prefer to clump together more tightly, forming loose aggregates or flocs. These particles exist as loose aggregates, showing a high sedimentation rate when they settle together as a population. Sediments formed in this manner are loosely anchored and often have a scaffold-like structure that is prone to resuspension [74]. In contrast, some researchers have observed that larger particles have stronger adjuvant effects. For example, a study conducted by Gutierro to capture BSA with PLGA particles showed that particles of size 1000 nm induced stronger serum IgG responses than particles of size 200 and 500 nm by intranasal, oral, or subcutaneous administration [75]. The same result was found by Kanchan et al. [76], that higher antibody titers are induced when administrating intramuscularly with 2–8 μm PLA particles loaded with hepatitis B surface antigen than compared with those sized 200–600 nm. It is reported by Mann et al. that influenza A antigen embedded in bilobal particles of size 980 nm induced stronger serum IgG2α responses by oral administration, producing more IFN-γ than 250 nm particles [77]. On the contrary, several other studies have demonstrated that large and small particles have a similar degree of stimulation effect. Wendorf et al. loaded env glycoprotein gp140 antigen from type B Neisseria meningitidis and from HIV-1 onto PLGA particles of different sizes and administered them intramuscularly, intraperitoneally, or intranasally, they identified that serum IgG levels between 110 nm and 800–900 nm particles and IgG1/IgG2 ratios with no significant differences [78]. By oral or subcutaneous administration, Gutierro et al. observed that PLGA particles of sizes 200 nm and 500 nm induced similar IgG, IgG1, and IgG2a responses [75]. Jung et al. noted by an intranasal application that 100 nm and 500 nm Poly (Vinyl Alcohol)-Graft-Poly (Lactide-coGlycolide) (PVAL-gPLGA) particles induced comparative levels of IgG and IgA [79]. Nagamoto et al. also showed that particles with different sizes resulted in similar IgG levels [80].

The particle size of the adjuvant is a decisive factor for adjuvants’ properties such as the available adsorption surface area for antigens. In turn, this also influences the conformation of antigen and may present functional epitopes to the immune system of the immunized individual. On the basis of experimental studies, the optimal particle size of AP is about 10 μm [81], closely matching the particle size distribution of approximately 11 μm, which is an average value reported by Mei et al. [82] in their study using a laser to test particle sizes. The particle size distribution is consistent with data collected over the years for AP batches, and the overall contours and derived diameters are the same as those previously reported by Kalbfleisch et al. [83].

### 5.5. Types of Antigens and the Doses Applied

There are several binding mechanisms between antigen and AP, among which two mechanisms, electrostatic attraction and ligand exchange, largely determine the adsorption and elution properties of vaccines [27]. Ligand exchange induced the strongest adsorption between the antigen and adjuvant. However, intensive adsorption can negatively affect the immunogenicity; for example, the substitution of hydroxyl groups in the adjuvant by phosphate groups in HBsAg leads to intensive adsorption, and the difficulty of vaccine desorption in vivo prevents the uptake of the antigen and T cell activation by APCs [9]. Antigen properties and chemical environment (pH, ionic strength, surfactant) affect the adsorption between antigen and adjuvant [71,84,85], and the surfactant increases the hydrophobic interaction between antigen–adjuvant and decreases the electrostatic adsorption between antigen–adjuvant. When rabbits were injected intramuscularly with Al^26^-labeled AH and AP, the bio-distribution sites in vivo were the same, but AP dissolved faster and had a higher concentration in the tissue fluid than AH, which was determined by the amorphous structure of AP [20].

After an intramuscular injection of vaccine, due to the complicated physiological environment in the interstitial fluid, the antigen–adjuvant that relies on electrostatic adsorption dissolves rapidly, and the antigen–adjuvant in the existence of ligand exchange dissolves more slowly, and the final dissolution products consist of an antigen, adjuvant, antigen–adjuvant and aluminum ions [86,87]. Researchers have used various types and doses of model antigens such as OVA [88], BSA [75], and HIV Tat [78] for the investigation of the effect of adjuvant particle properties on the immune response, triggering a controversy about the size-dependent immune response. Different antigens have variable purities, innate immunogenicities, etc., and therefore, the characteristics of adjuvant particles have a unique effect on the immune response from one antigen to another. In some extreme cases, multiple doses of highly immunogenic antigens can result in similar antibody levels, thus reducing the influence of the adjuvant particle size and other properties on humoral immunity [89]. The pH of the adjuvant surface is often different from the native pH. According to the Gouy–Chapman bilayer theory, the adsorbed antigen will undergo chemical degradation reactions at a rate related to the pH of the microenvironment rather than the pH of proprioception. In an aqueous medium, the surface charge on a suspending particle affects the ion distribution of the surrounding solvent. Negatively charged particles will prefer to attract cations (including protons) into the double layer surrounding the particle. Positively charged particles preferentially tend to attract anions (including hydroxyl groups), which enter the bilayer [4]. The adsorbed antigen is exposed to the pH environment of the bilayer, and any pH-dependent reaction occurs at a rate related to the pH of the bilayer rather than to the native pH [90]. Therefore, the efficacy of a vaccine containing AP also depends on the properties of the antigen presented in the vaccine and the formulation. The properties of antigens that should be characterized in pre-formulation studies include as follows: PZC, the existence of accessible phosphate groups or groups generating phosphate groups during aging process, and the pH effect of the solutions on chemical and conformational stability.

## 6. Safety Profiles and Potential Adverse Effects from AP

The mechanisms of action of AH and AP are essentially the same and they share similar safety concerns. The main differences between AH and AP lie in their chemical properties and ability to adsorb antigens. From an immunological perspective, aluminum-containing adjuvants induce IgE-mediated hypersensitivity reactions. Aluminum-containing adjuvants can cause inflammation at the injection site and stimulate local erythema and granulomas. Aluminum-containing adjuvants do not effectively enhance the Th1-type immune response, as they interfere with cellular immunity and block the activation of CD8^+^ T cells, resulting in incomplete and non-persistent immunity of vaccines [91]. For example, Bacard et al. [92] presented the results of preclinical safety testing of Quimi-Hib vaccine with aluminum phosphate adjuvant and found that sclerosis caused by AP-induced macrophage granulomas was detected at the vaccination site. Our previous studies have summarized in detail the disadvantages and side effects of AH [37]. Additional studies have shown that most of the aluminum is rapidly cleared by the kidneys after vaccination [93]. However, infants, the elderly and those with impaired kidney function are prone to accumulation of aluminum in their bodies [94]. After entering the human body, aluminum-containing adjuvants first deposit in the brain, producing neurotoxic effects [95], which are manifested in memory loss, inattention, language dysfunction and behavior changes, and can also be manifested in psychomotor disorder, repetitive behavior, language disorder, sleep disorder, seizure, anxiety and inattention, learning and memory defects. In addition, Alzheimer’s disease, Parkinson’s disease, and multiple sclerosis may all be associated with aluminum accumulation toxicity [96,97]. According to research data evaluation, the concentration of aluminum in the blood produced by vaccines administered to one-year-old infants is still much lower than the toxic substances and the prescribed minimum risk level [98].

A 100% risk-free vaccine does not exist, but there is insufficient evidence to confirm the toxicity of aluminum-containing adjuvants. The risk of serious and rapid adverse reactions during the use of aluminum-containing adjuvants is very low, so we should have a high level of confidence in the safety of existing vaccines containing aluminum adjuvants.

## 7. The Status of AP Improvement

Generally, the aluminum adjuvant is effective in increasing serum antibodies, while its ability to assist in the induction of cellular immunity is weak and can cause injection site reactions, which limits its application to some extent. From the perspective of immunology, the aluminum adjuvant lacks an adjuvant effect or has a weak adjuvant effect on some antigens, and cannot induce cellular immunity and cytotoxic T cell response while enhancing humoral immunity. In addition, aluminum-containing adjuvants can induce IgE-mediated hypersensitivity reactions. In general, aluminum adjuvants can enhance type II immune responses and stimulate the production of cytokines such as IL-4 and IL-5, but do not induce type I immune responses. Aluminum salt can cause an inflammatory reaction at the injection site, stimulating local erythema, granuloma, and subcutaneous nodules. For these reasons, regulatory authorities have set upper limits for the number of aluminum-containing adjuvants in human vaccines, which may affect the efficacy of certain aluminum adjuvanted vaccines [99], and AP plays a key role in inactivated human vaccines and subunit prophylactic vaccines. Therefore, there is a long way to go to improve AP or to combine AP with other adjuvants to form complex adjuvants and to broaden the application of AP in vaccines.

### 7.1. Improvement of AP Formulation

An amorphous aluminum hydroxyphosphate sulfate adjuvant (AAHS) is a sulfate-containing AP manufactured by Merck, which is prepared by precipitating aluminum potassium sulfate with sodium hydroxide [100]. The recent development by Merck has applied AAHS as preferential inclusion or AP replacement in many vaccine formulations, such as quadrivalent and nine-valent HPV vaccines (Gardasil-4 and Gardasil-9), Hib vaccine (PedVaxHIB), encephalitis A vaccine (VAQTA) and encephalitis B vaccine (RecombivaxHB) [101]. At the macromolecular level, AAHS is amorphously reticulated, and AAHS adjuvants have a P/Al ratio of 0.3 and a zero charge point of approximately 7.0, which is close to zero charges in neutral solutions, whereas both AP and AH are strongly charged at neutral pH. The net charge of the adjuvants may have an influence on their ability to bind the antigen, and also affect the antigen after injection into the host and exposure to a serum because the serum of the host may be close to neutral pH. Caulfield et al. [100] showed that the adsorption capacity of AAHS to L1 VLPs was enhanced compared to both AH and AP, and AAHS was essentially 100% of the vaccine. AAHS adjuvant triggered high levels of HPV-specific IgG1 antibodies through intramuscular administration in mice, but these responses had no statistically significant difference from the AP-enhanced responses. The characterization of this adjuvant is still at a preliminary stage and there are few related literature sources, making it difficult to give an objective evaluation of the efficacy and safety of this preparation with respect to its structural characteristics.

### 7.2. Nano-AP

On the premise of ensuring the safety and effectiveness of an aluminum adjuvant, it is feasible and practical to enhance its adjuvant effect by changing its particle size and shape, starting from the mechanism of adjuvant effect. Researchers have found that nano-AP can modulate the immune effect of vaccines by changing its particle size and surface chemistry. For example, modulating the surface charge and hydroxyl content of nano-AP can increase its binding affinity to antigens and improve the immunogenicity of vaccines [102]. The preparation of nano-AP depends on the conditions of preparation, including the concentration of reagents, the order of the adding of reagents, the rate of the adding and mixing of reagents, the stirring rate, the pH of the precipitation process, etc. The PZC and particle size of the resulting AP can be precisely regulated by changing the pH during the reaction process, which is based on the principle that as the pH of the interaction rises, the PO_4_/OH ratio of the final product Al(OH)x(PO_4_)y rises as well, and the content of phosphate-substituted hydroxyl groups increases and the PZC of the product decreases, therefore, the preparation of aluminum phosphate nano-adjuvants can be achieved by controlling the conditions of adjuvant preparation. Raponi et al. [103] have clearly verified that the immunological advantage can be implemented by reducing the size of aluminum adjuvant particles to produce a more effective adjuvant stimulating cellular and humoral immune responses. The mechanistic basis for this improved immune performance is currently unknown, but the behavior of other nanoparticles in biological systems may play an essential role in delivery effects, such as improving the distribution of draining lymph nodes, increasing antigen uptake, and improving the size and duration of antigen presentation [104,105]. Aluminum nano-phosphate adjuvants (AP-NPs) exhibited stronger adjuvant activity than conventional aluminum phosphate adjuvants (AP-MPs) in mouse models and were more effective than AP-MPs in activating NLRP3 inflammatory vesicles, and AP-NPs stimulated THP-1 cells to produce higher levels of IL-1 than AP-MPs [60]. Nano-sized aluminum-containing adjuvants with smaller particle size, increased specific surface area, enhancing binding area to antigen, and better biocompatibility can elevate the immune response of the body and drastically decrease the side effects of adjuvants, providing a novel and promising research route for the development of vaccines important for Th1 immunity, including vaccines against infectious diseases such as tuberculosis, pertussis, and malaria or cancer [106,107]. He et al. [37] have summarized the research progress of nano-AH. However, due to the differences in immune mechanisms between AP and AH, the application of nano-AP in vaccine preparation is receiving increasing attention.

Liang et al. [108] synthesized amorphous aluminum hydroxyphosphate nanoparticles (AAHPs) with good physicochemical properties by chemical precipitation and explored the mechanism of specific interaction of AAHPs with cells using a recombinant protein antigen of *Staphylococcus aureus* (*S. aureus*) and human papillomavirus (HPV) type 18 virus-like particles (VLP) as model antigens. It was shown that AAHPs can cause cell membrane perturbation accompanied by potassium efflux and can further promote cellular internalization. Downstream inflammatory responses were observed after cellular uptake, as evidenced by the production of mitochondrial ROS and the production of pro-inflammatory cytokines that mediate lysosomal damage via particle-phosphate lipid interactions. When binding AAHPs with *S. aureus* recombinant protein antigens or HPV type 18 virus-like particles (VLPs), the surface charge of AAHPs plays a dominant role in regulating adjuvant properties by inducing a robust and persistent antigen-specific humoral immune response. The above findings reveal potential immunostimulatory mechanisms of nano-aluminum phosphate and further provide strategies for the design of nanomaterial-based adjuvants in prophylactic and therapeutic vaccines.

Although nano-aluminum-containing adjuvants, including nano-AP, have many potential advantages, no aluminum nano-adjuvants have been approved for human use so far, and their safety and toxicity are also important issues of concern for researchers. Therefore, before nano-AP enters clinical trials, it is necessary to conduct systematic safety assessments and toxicological studies on nano-AP to ensure its safe application in vaccines.

### 7.3. Composite Adjuvant Containing AP

In order to induce both type I and type II immune responses simultaneously, it is necessary to add additional components to the aluminum adjuvant in order to exert its optimal effect. Combining immunostimulatory substances targeting different activation pathways for immune response with aluminum-containing adjuvants effects synergistically, potentially leading to a more effective or longer-lasting immune response and a reduction in the antigen content used. Aluminum-containing adjuvants induce only weak Th1 and Th17 responses, yet Th1 and Th17 responses may be required to induce protective immunity against certain important infectious diseases such as malaria and tuberculosis [102]. The adsorption of immune-stimulating molecules on aluminum-containing adjuvants limits their systemic distribution, thereby reducing the risk of systemic side effects, enhancing the targeting to antigen-presenting cells of these molecules and co-adsorbed antigens. Ligands for pattern recognition receptors, especially Toll-like receptors (TLRs), are outstanding candidates for combinatorial adjuvants because they are localized within the cell membrane or intracellular compartments; besides, they signal through the MyD88 and TRIF pathways, which are complementary to the cellular signaling pathways activated by aluminum-containing adjuvants [109]. For example, the AS04 adjuvant composed of aluminum adjuvant and TLR4 agonist monophosphate A (MPL) is the first approved combinatorial adjuvant for human vaccines (including HPV and hepatitis B vaccines). The AS04 adjuvant used in the hepatitis B vaccine (Fendrix), which is mainly used to prevent hepatitis B in hemodialysis patients, is a combination of AP and MPL [110]. MPL stimulation activates the innate immune response and also promotes IFN-γ production via CD4^+^ T cells, biasing the immune response toward Th1 [111]. Aluminum-containing adjuvants can bias the immune response towards Th2. The combination of aluminum-containing adjuvant and MPL adjuvant achieves a dual effect [112]. The AS04 adjuvant effectively activates different branches of the immune response, and not only helps in inducing higher antibody titers, but most importantly, helps improve the retention time of circulating antibodies [113], mechanism of action of the AS04 adjuvant are summarized in Figure 2.

Other Toll-like receptor antagonists are in the process of development, or have already been marketed, such as CpG oligonucleotides [114]. In addition, the combination of aluminum adjuvant and IL-18 [115] may also result in a promising immune effect. For example, with the development of cancer vaccines, the elements of powerful immunogenic cancer-specific antigens and strong adjuvants to stimulate enhanced cellular immunity to clear cancer cells should be included at the same time for an ideal cancer vaccine. However, most commercially available adjuvants fail to stimulate a strong cellular immune response. Gan et al. modified AP to nanoscale and prepared a CpG-loaded, B16F10 tumor cell membrane-encapsulated aluminum phosphate nanoparticles (APMC), which are approximately 60 nm in size, as a lymph-node-targeted cancer vaccine. In addition to acting as a tumor antigen, the tumor cell membranes can also efficiently improve the colloidal dispersion of aluminum phosphate nanoparticles. Studies have proved that subcutaneous injection of APMC effectively drains into mouse lymph nodes, significantly enhancing the co-internalization of tumor antigens and CpG into antigen-presenting cells (APCs) and further promoting cell maturation. After immunization of mice, they induced high-intensity cellular immunity, including potent IFN-γ^+^ CD4^+^ T cells, IFN-γ^+^ CD8^+^ T cells, cytotoxic T lymphocytes, and cytokine secretions in spleen and lymph node cells. The significant inhibition of tumor growth and the survival prolongation were observed in mice from melanoma models both preventive and therapeutic [115].

## 8. Key Points of Preclinical and Clinical Studies on AP Improvement

The effect of adjuvants on the immunogenicity of vaccines: (1) Firstly, it needs to be proved that the addition of the adjuvant can indeed induce a long-term and efficient specific immune response and improve the protective ability of the organism more effectively than conventional AP, which is the basis for the application of the adjuvant. If the study results show that the addition of an adjuvant has little effect on the enhancement of immunogenicity, it should not be added to avoid side effects caused by the adjuvant. Immunological effects should include both humoral and cellular immunity. In addition to functional antibodies (neutralizing, modulating phagocytic or bactericidal antibodies), the detection of humoral immunity should also include subclasses of immunoglobulins. At the same time, detection of cellular immunity should include antigen-specific T-cell responses, including Th1, Th2, T-cell regulatory factors and/or related cytokines, etc. (2) Toxicology and pharmacology of adjuvants: In terms of the toxicology of adjuvants, the main considerations are the pathological response at the vaccination site, the antibody response produced by the organism, and the duration of antibodies. Aluminum-containing adjuvants usually delay the production of antibodies against the vaccine and increase the incidence of local side effects of the injection. If the amount of antigen in the vaccine can meet the needs of immunization, it is recommended that no adjuvant be added. The pharmacological test of the adjuvant should mainly consider the principle of action, the relationship between the result of action and the dose of the vaccine, the immunization procedure and route of vaccination and the immunization effect, etc. (3) Interaction between adjuvant and other components of vaccine: As the adjuvant is an important vaccine component, all components in the adjuvant should be compatible with other vaccine components, such as preservatives, inactive components, etc. The influence of adjuvants on the adsorption of different components should be considered, including whether there is dissociation after adsorption and the degree of dissociation, as well as the effect of the addition order of each component during the adsorption process. (4) Metabolism of adjuvants when administered alone: It is necessary to have sufficient preclinical safety data to support the study, and considering that the adjuvant may accumulate in the body. Additionally, human pharmacokinetic studies should be conducted when adjuvants are administered alone. The design of human pharmacokinetic studies should be based on preclinical study data and conducted according to the properties and characteristics of the adjuvant in order to obtain scientific and convincing results. (5) Dose of adjuvant: The dose of adjuvant used and the ratio of the adjuvant to the antigen are related to whether the desired immune response can be induced and the adverse effects must be minimized. It is necessary to refer to the dosage of similar marketed products, and attention should be paid to both the dosage of adjuvant and the selection of the optimal ratio of adjuvant to antigen. Studies should be conducted in the target population as much as possible and multiple dose groups should be designed. (6) Immune-boosting effect: Clinical observations should follow the GCP principles and the design should not only be randomized, double-blind, and controlled, but should also combine the characteristics of antigens and adjuvants. Observations should be made in the target population, and if the age of the target group spans a wide range, prior stratification or more than one clinical observation should be considered. To evaluate its protection effect, it is necessary to use marketed vaccines that can prevent the same disease as a control. If not available, other unrelated vaccines available to the target population may also be used as placebos. If necessary, multiple controls should also be established. The sample volume should meet the minimum statistical requirements. For non-inferiority tests, the threshold for non-inferiority must be set in advance and the basis for setting it must be provided. In designing the analytical methods and sample size for clinical observation, appropriate consideration should also be given to the influence of multiple factors. (7) Local and systemic reactions: Studies have shown that absolutely safe adjuvants do not exist. It is only possible to adjust the adjuvant according to its mechanism of action to maximize the immunostimulatory effect and minimize the toxic side effects as much as possible. Clinical observations should be designed with full consideration of the collection of adverse reactions caused by the vaccine. There should be a comparison and analysis of adverse reactions with vaccines (adjuvants and antigens) that have been marketed, and a systematic and detailed collection of adverse reactions, including local and systemic reactions, laboratory test indices of relevant systems, and instrumentation results, in accordance with relevant national requirements.

## 9. Summary

Aluminum-containing adjuvants have been proven safe and effective over decades of vast population use. We remain convinced that aluminum-containing adjuvants remain a key benchmark and the “potency ruler”. Although newer adjuvants may and will prove to be more effective than aluminum, specific critical assessments of enhanced efficacy are needed, and the level and type of immune response required for protective immunity against a particular disease cannot be fully determined and will need to be established in large clinical studies. In many programs, the development time, problems in production and procurement of raw materials and excipients, the cumulative data required, and the regulation involved would be greatly reduced if aluminum-based adjuvants were chosen for product development. Scientists who are keen to promote more new approaches should not ignore the intrinsic value of having an aluminum adjuvanted vaccine product on the market. Most current studies on the mechanism of action of aluminum-containing adjuvants have used AH adjuvant as the research object, and fewer studies have been conducted on AP adjuvants [116]. However, previous studies have shown that the mechanisms in the stimulation of immune response and factors affecting the immunogenicity of aluminum-containing adjuvants have certain similarities, but there are also certain differences.

Recent studies are more focused on improving our realization on the mechanisms of antigen adsorption by AH and AP adjuvants and the effects of adsorption on antigen stability and immune responses as well. To specifically understand how aluminum-containing adjuvants enhance immune response through several molecular pathways, it is necessary to conduct highly interdisciplinary research in fields such as molecular and cellular immunology, intracellular transport, biomaterial chemistry, toxicology, and pathology. Such kinds of knowledge will do us a favor to decide the optimal simulating conditions for an effective and safe aluminum adjuvant vaccine to study the interactions and relative importance of these pathways, including (1) molecular level research to elucidate the cell activation mechanisms mediated by aluminum adjuvant; (2) cellular level studies to elucidate the mechanisms of uptake of the antigen-loaded aluminum adjuvant by immune cells and the activation and migration of these cells; and (3) in vivo studies that emphasize the underlying mechanisms controlling immune regulation [117], to further understand the differences in immune mechanisms between the two aluminum-containing adjuvants [34]. However, aluminum-containing adjuvants also have certain limitations, such as the induction of severe local tissue irritation, prolonged inflammatory response accompanying the injected site, stronger Th2 immune response, poor ability to induce cell-mediated immune response, tendency to produce IgE response, etc. In addition, aluminum adjuvanted vaccines usually fail to induce antiviral immunity [118]. Since aluminum-containing adjuvants are still a key benchmark for judging the safety and efficacy of other adjuvants, the improvement of aluminum-containing adjuvants or the combination of aluminum-containing adjuvants with other adjuvants to form complex adjuvants and the broadening of the application of aluminum-containing adjuvants in vaccines are important directions of adjuvant research and have yielded certain results, such as AAHS adjuvants, nano-aluminum-containing adjuvants and AS04 adjuvants. With the development of a new generation of antigens, researchers need to further optimize the routes to design novel applications of more effective adjuvants in different antigen vaccine formulations by utilizing the interdisciplinary expertise mentioned above, and comprehensively evaluate their safety. However, in order to create a new generation, what has been mentioned in this review has become particularly important.

## Figures and Tables

**Figure 1 pharmaceutics-15-01756-f001:**
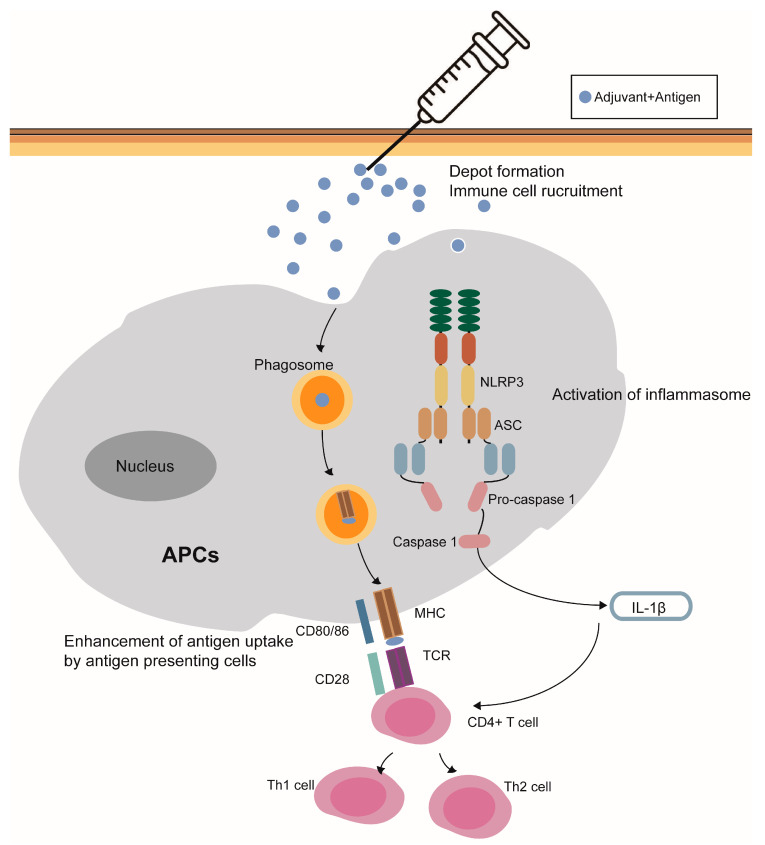
Modes of action of aluminum-containing adjuvants. (1) Depot effect; (2) recruitment of immune cells; (3) enhancement of antigen uptake by antigen presenting cells; (4) the NLRP3 inflammasome activation; and (5) potentiality to stimulate the humoral immune responses.

**Figure 2 pharmaceutics-15-01756-f002:**
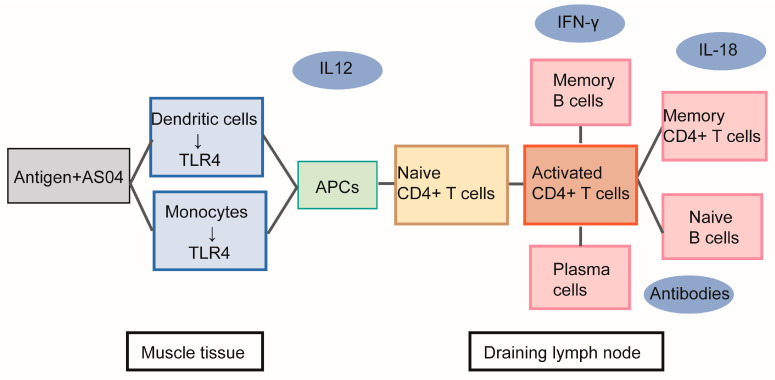
Mechanism of action of the AS04 adjuvant.

**Table 1 pharmaceutics-15-01756-t001:** List of licensed type of adjuvant and aluminum content of aluminum-adjuvanted vaccines for use in the United States (Information accessed on FDA website on 5 May 2023 (https://www.fda.gov/vaccines-blood-biologics/vaccines/vaccines-licensed-use-united-states, accessed on 5 May 2023)).

Adjuvant	Vaccine	Tradename	Manufacturer	Dose (Al^3+^)
AH	Anthrax	BioThrax^®^	Emergent Biosolutions	1.2 mg
AH		Infanrix^®^	GSK	≤0.625 mg
AH	DTaP, polio	Kinrix^®^	GSK	≤0.6 mg
AH	DTaP, Hepatitis B, polio	Pediarix^®^	GSK	≤0.85 mg
AH	Hepatitis A	Havrix^®^	GSK	0.5 mg (adult) 0.25 mg (pediatric)
AH	Hepatitis B	Engerix	GSK	0.5 mg (adult) 0.25 mg (pediatric)
AH		PREHEVBRIO^®^	VBI	0.5 mg
AH	Human papilloma virus	Cervarix^®^	GSK	0.5 mg (plus 50 µg MPLA)
AH	Meningococcus B	Bexsero^®^	GSK	0.519 mg
AH		Boostrix^®^	GSK	≤0.39 mg
AH	Tick-Borne Encephalitis	TICOVAC^®^	Pfizer	0.35 mg (adult) 0.175 mg (pediatric)
AH	Japanese Encephalitis	IXIARO^®^	Valneva Austria GmbH	0.25 mg
AH and AP	Hepatitis A and Hepatitis B	Twinrix^®^	GSK	0.45 mg
AP	Diphtheria and Tetanus Toxoids Adsorbed	/	Sanofi-Pasteur	0.33 mg
AP		/	Sanofi-Pasteur	0.33 mg
AP		Tenivac^®^	MassBiologics	≤0.53 mg
AP		TDVAX^®^	MassBiologics	≤0.53 mg
AP	DTaP	Daptacel^®^	Sanofi-Pasteur	0.33 mg
AP		Adacel^®^	Sanofi-Pasteur	0.33 mg
AP	DTaP, polio	Quadracel^®^	Sanofi-Pasteur	0.33 mg
AP	DTaP, polio, Hib	Pentacel^®^	Sanofi-Pasteur	0.33 mg
AP	DTaP, polio, Hib, Hepatitis B	VAXELIS^®^	MSP	0.319 mg
AP	Meningococcus B	Trumenba^®^	Pfizer	0.25 mg
AP	Meningococcus B	TRUMENBA^®^	Wyeth	0.25 mg
AP	Pneumococcus 13-valent Conjugate	Prevnar 13^®^	Pfizer	0.125 mg
AP	Pneumococcal 15-valent Conjugate	VAXNEUVANCE^®^	Merck	0.125 mg
AP	Pneumococcal 20-valent Conjugate	PREVNAR 20^®^	Pfizer	0.125 mg
AAHS	Hib	PedVaxHIB^®^	Merck	0.225 mg
AAHS	Hepatitis A	VAQTA^®^	Merck	0.45 mg (adult) 0.225 mg (pediatric)
AAHS	Hepatitis B	Recombivax HB^®^	Merck	0.5 mg (adult) 0.25 mg (pediatric)
AAHS	Human papilloma virus	Gardasil^®^	Merck	0.225 mg
AAHS	Human papilloma virus	Gardasil-9^®^	Merck	0.5 mg

AH: aluminum hydroxide adjuvant; AP: aluminum phosphate adjuvant; AAHS: amorphous aluminum hydroxyphosphate sulfate; DTaP: diphtheria toxoid, tetanus toxoid, acellular pertussis; Hib: Haemophilus influenza B.

## Data Availability

No new data were created or analyzed in this study. Data sharing is not applicable to this article.

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
