# Peer review of "Research Progress of Aluminum Phosphate Adjuvants and Their Action Mechanisms"

_pharmaceutics, 2023, doi:10.3390/pharmaceutics15061756_

Round 1

Reviewer 1 Report

The submitted review “Research progress of aluminum phosphate adjuvants and their action mechanisms”, thoroughly describe the application of aluminum-based adjuvants within vaccine formulations in terms of immunogenicity, physiochemical characteristics, as well as status of improvements towards forthcoming adjuvant generations. This manuscript is comprehensive, well-structured, and could be considered a valuable guidance for future development and optimization. Suggestions and comments are to be addressed prior publication:

1. Authors should provide brief literature data comparing the application of different vaccine formulation additives (e.g. aluminum versus saponin and others), highlighting the advent of employing aluminum-based adjuvant over others.

2. “A picture is worth a thousand words”, providing schematic diagrams capable of summarizing lots of information is always considered relevant for any manuscript as well as attractive to the readers for better data comprehension. Therefore, providing a schematic diagram that highlights the current evidence mechanistic aspects of aluminum-based adjuvants immunogenicity is highly recommended. Additionally, sketched diagram for formulation improvement would be relevant.

3. Although aluminum adjuvants are typically thought to induce cell-based immune responses, recent evidence showed the aluminum adjuvant potentiality to mount different immune responses; humoral one (please refer to Evaluation of the Association of Recombinant Proteins NanH and PknG from Corynebacterium pseudotuberculosis Using Different Adjuvants as a Recombinant Vaccine in Mice”, Vaccines 2023, 11(3), 519; https://doi.org/10.3390/vaccines11030519).

4. Safety profiles and potential adverse effects from aluminum-based adjuvants should be highlighted within a separate section.

5. Providing a section for analytical characterization of aluminum-adsorbed antigens as a mean of quality control would also be relevant.

Reviewer 2 Report

Research progress of aluminum phosphate adjuvants and their  action mechanisms is a through and interesting examination of the adjuvant properties of aluminum and the potential continued use of such adjuvants in the future, if only to serve as a benchmark for efficacy of newer materials. 

The only issue with the paper is some clumsy writing and some minor grammar issues.  The first sentence of the introduction should be rewritten. 

Line 85”convincing” –convinced

Line 129 resulted—resulting

Line 175-improve—improved

Line 195 amount-number

ON line 256, please explain the meaning of “transport-related pathway”

Line 268 no “down” regulation?

Line 288 recruit “white” blood cells ?

There are probably a few more places that need refinement.  Please, go over the paper once more to check grammar and word selection.

 After the first sentence of the introduction, which is horrible, the rest is fairly good with a few minor issues.

Round 2

Reviewer 1 Report

The authors kindly responded to all comments and suggestions. 

Author Response

We deeply appreciate your consideration of our manuscript.
Thanks for your attention.